Dragonflies and damselflies (Odonata) from Puerto Rico: a checklist with notes on distribution and habitat

Ramírez Alonso alonso.ramirez@ncsu.edu 1
Maldonado-Benítez Norman 2
Mariani-Ríos Ashley 2
Figueroa-Santiago Javier 3
1 Department of Applied Ecology, North Carolina State University , Raleigh , NC , United States of America
2 Department of Environmental Sciences, University of Puerto Rico , San Juan , Puerto Rico
3 Resource Center for Science and Engineering, University of Puerto Rico , San Juan , Puerto Rico
Gillespie Joseph
Electronic publication date: 2020 Oct 1
Publication date: 2020
Volume: 8
Electronic Location ID: e9711
Received 2020 May 27; Accepted 2020 Jul 23
Copyright: ©2020 Ramírez et al.
Copyright year: 2020
Copyright holder: Ramírez et al.
License: This is an open access article distributed under the terms of the Creative Commons Attribution License, which permits unrestricted use, distribution, reproduction and adaptation in any medium and for any purpose provided that it is properly attributed. For attribution, the original author(s), title, publication source (PeerJ) and either DOI or URL of the article must be cited.
License URL: https://creativecommons.org/licenses/by/4.0/

Keywords: Species inventory, West Indies, Insect diversity, Caribbean

Funding: Puerto Rico Louis Stokes Alliance for Minority Participation National Science Foundation This work was supported by the Puerto Rico Louis Stokes Alliance for Minority Participation, National Science Foundation. The funders had no role in study design, data collection and analysis, decision to publish, or preparation of the manuscript.

==============================
Background

Conservation of tropical freshwater fauna requires a solid understanding of species biodiversity patterns. We provide an up to date annotated list of Odonata of Puerto Rico, which is based on current reports. The list is complemented with notes on the geographic and altitudinal distribution of this order on the island. We also compare current composition relative to early reports conducted when Puerto Rico was mostly an agricultural region.

Methods

We surveyed adult Odonata all over Puerto Rico with the aid of undergraduate students. Students were trained on capturing, preserving, and data basing specimens. All material was centralized, identified by the lead author, and deposited in the Zoology Museum at the University of Puerto Rico (MZUPR), Río Piedras campus. Surveys were complemented with focal collections by the authors and a literature review of published records for Puerto Rico and the Caribbean. We requested records from specialists to obtain the most complete list of species for the island.

Results

An annotated list of Odonata from Puerto Rico is presented, reporting 49 species distributed in two suborders and four families. We provide information on species distribution among municipalities and elevations around Puerto Rico. A historic list of species was developed for the 1930s-1940s, when agriculture covered most of Puerto Rico, based on literature and museum specimens. Both current and historic records are similar and suggest that the Odonata fauna is dominated by generalist species and has changed little since the agricultural period. Our list provides a point of reference to understand biodiversity patterns in Puerto Rico and the Caribbean and for assessing the effects of land use change on aquatic insect diversity.

Introduction

Tropical freshwater biodiversity faces a wide variety of threats, ranging from a globally changing climate to point source discharges of pollutants (Ríos-Touma & Ramírez, 2018). The combination of these stressors is changing aquatic ecosystems, altering their function, and decreasing and homogenizing freshwater biodiversity. Understanding the effects of multiple stressors is critical for the proper management and restoration of ecosystems. Information on native biodiversity is a critical component to understand and mitigate impacts on freshwaters. Although our information on tropical diversity has increased considerable, we still have major gaps for many groups of freshwater species (e.g., aquatic dipterans and coleopterans).

Dragonflies and damselflies, order Odonata, are one the most ancient and charismatic groups of insects. There are approximately 6,338 species reported globally, but scientists agree there are many more to be discovered (Paulson & Schorr, 2020). Adults have bright coloration, are active fliers, and relatively easy to observe in their natural habitats, making them attractive to humans, and useful in research studies (e.g., bioassessment). They are also important components of freshwater ecosystems, where they are predators. Adults consume a variety of flying insects and even spiders, while larvae consume small invertebrates and small vertebrates (e.g., fish, tadpoles). Our knowledge of Odonata diversity in the tropics is still limited, but perhaps better than that of other freshwater groups. Thus, the order could be used as indicator of how anthropogenic activities affect freshwater biodiversity (Clausnitzer et al., 2009).

Puerto Rico provides an interesting case study to understand the effects of land use changes in tropical biodiversity. Deforestation and forest conversion into agriculture followed the common pattern observed in most of the tropics. By the late 1940s, only 6% of the island had some sort of forest cover (Helmer, 2004). In the 1940s, Puerto Rico changed its economic model for an industrialized one, stimulating rural to urban migration, and abandonment of agriculture (Grau et al., 2003). Industrialization resulted in a steady increase in secondary forest and urbanization until the 2000s when forest cover values became stable at 40% (Grau et al., 2003). This same trend is now observed in different parts of Latin America, where forest regrowth is occurring following changes policies and economic models (Aide et al., 2013). Initial Odonata surveys in Puerto Rico were conducted during the period of intense agricultural activity, from 1930s to 1940s (Klots, 1932; García Díaz, 1938). Thus, we have an opportunity to assess changes in freshwater biodiversity in response to increased forest cover.

In this study, we provide an up to date annotated list of Odonata of Puerto Rico based on recent reports and a survey of the island. Studies on Odonata of Puerto Rico are limited, with reports from the 1800s and early 1900s, during the peak in agricultural activity on the island. Klots (1932) published the first list of Odonata for the island, followed by a list by García-Díaz in 1938. Together, the two studies reported 39 species of Odonata for Puerto Rico. Sporadic studies that included Puerto Rico increased this number and confirmed some reports (e.g., Garrison, 1986). Meurgey (2013) reviewed early reports for the Caribbean and questioned the validity of some, including several for Puerto Rico, and decreased the number of species for the island to 36. The change was mostly due to errors in locality information provided by early collectors (Meurgey, 2013). Our goal is to provide a comprehensive list of Odonata species for Puerto Rico with notes on their distribution (e.g., municipalities, elevation). We assess changes in composition relative to early reports during the agricultural peak in Puerto Rico. Overall, we expect our study to advance our understanding freshwater biodiversity in oceanic islands and aid in understanding how land use changes affect freshwater biodiversity in the tropics.

Materials & Methods

Puerto Rico is the smallest of the Greater Antilles in the Caribbean, with 8,900 km2 and elevations that range from sea level to 1,338 m. The climate is tropical with a strong marine influence, thus air temperatures range from 25 to 31 °C and precipitation from 1,000 to 4,000 mm per year. A dominant eastern wind interacts with the central mountain range to create a rain shadow over the southwest part of the island, where the climate is strongly seasonal and drier than the remaining of the area (Ramírez & Gutiérrez-Fonseca, 2014).

Puerto Rico has a diversity of life zones, including subtropical moist forest, subtropical wet forest, and subtropical dry forest (Ewel & Whitmore, 1973). The island is divided into 134 hydrological units distributed into 64 watersheds that discharge directly to the ocean, 10 of these watersheds are intermittent. In addition, there are over 70 coastal wetlands with limited water flow and strong interactions with the aquifer. The Cartagena lagoon is the only natural lake on the island, located on the southwestern side of the island (Ramírez & Gutiérrez-Fonseca, 2014).

We surveyed adult Odonata all over Puerto Rico using two main approaches (Fig. 1). The lead author has been sampling the island since 2001, conducting surveys at different locations over time. Then, during 2015–2016, with support from the Puerto Rico Louis Stokes Alliance for Minority Participation program, sponsored by the National Science Foundation, we created the Dragonfly Project (https://prlsamp.rcse.upr.edu/index.php/home/dragonfly-project). For this project, we divided the island into eight equally sized sectors and assigned a group of undergraduate students to each. Students were trained on specimen collection, handling and preservation, and data basing. We conducted two workshops per academic semester, for a total of 8 workshops during the project. Each group organized at least three sampling campaigns per semester within their designated areas. All material was centralized, identified by the lead author, and deposited in the Zoology Museum at the University of Puerto Rico (MZUPR), Río Piedras campus. Odonata surveys were complemented with a literature review of published records for Puerto Rico and the Caribbean. In addition, Rosser Garrison and Natalia von Ellenrieder were consulted to complete the list of species for the island.

Figure 1 Map of Puerto Rico.

Map of Puerto Rico showing the political boundaries of municipalities and major rivers. Each dot represents a sampling location where our student teams collected specimens for the survey.

We curated and identified 154 specimens deposited in the MZUPR by Prof. Julio García Díaz. The material dated from 1935 to 1947 and was stored in paper envelopes in boxes. We used this material in combination with published reports in the literature (Klots, 1932; García Díaz, 1938) to generate a historic list of species for Puerto Rico that coincides with the peak of agriculture in the island.

The checklist includes information on the first reliable literature record for each species. For species without published records for Puerto Rico, we provide the name of the person that first reported its occurrence.

Results

We found 49 species of Odonata in Puerto Rico, from four families (both suborders) and 29 genera (Table 1). Our team of students collected 2,451 specimens around the island, resulting in 33 species. Focal sampling by the lead author resulted in eight species not collected by students. Five additional species were added based on the literature and three more based on records provided by Rosser Garrison (Table 2). Our list omits two species previously reported in the literature as present in Puerto Rico that are now considered erroneous: Ischnura capreolus (Coenagrionidae; Meurgey, 2013) and Triacanthagyna trifida (Aeshnidae; Von Ellenrieder & Garrison, 2003).

Table 1 Species information: distribution and habitat.

List of species for Puerto Rico with information on their occurrence: municipality, elevation range, environment sampled, and land use.

			Species	Municipality	Elevation	Environment sampled	Land use	
Anisoptera					
	Aeshnidae					
		Anax					
			amazili (Burmeister, 1839)	San Juan	20	Open area	Urban	
			junius (Drury, 1773)	Cayey, Manatí, San Juan, Vega Baja				
		Coryphaeschna					
			adnexa (Hagen, 1861)	San Juana	20			
			viriditas Calvert, 1952	Cabo Rojoe, Quebradillas	200		Grassland	
		Gynacantha					
			nervosa Rambur, 1842	San Juan, Mayagüez	2-100	Canal	Urban	
		Rhionaeschna					
			psilus (Calvert, 1947)	Orocovisb	800	Open area	Road	
		Triacanthagyna					
			caribbea (Williamson, 1923)	Mayagüez	22	Stream	Forest	
			septima (Selys in Sagra, 1857)	Mayagüez	257	Stream	Grassland	
	Libellulidae					
		Brachymesia					
			furcata (Hagen, 1861)	Camuy, Juana Díaz, Ponce	71-280	Pond, Lake, Reservoir	Agriculture, Grassland, Urban	
			herbida (Gundlach, 1889)	Añasco, Arecibo, Cabo Rojo, Camuy, Carolina, Coamo, Fajardo, Guayanilla, Juana Díaz, Ponce	0-130	Pond, Lake / Reservoir, Mangrove, Stream	Agriculture, Grassland, Urban	
		Crocothemis					
			servilia (Drury, 1773)	Yabucoac	0-10	Pond	Beach-Coastal	
		Dythemis					
			rufinervis (Burmeister, 1839)	Adjuntas, Aibonito, Ciales, Cidra, Corozal, Guayanilla, Juana Díaz, Lares, Las Marías, Maricao, Mayagüez, Morovis, Peñuelas, Ponce, Río Grande, Salinas, Utuado, Villalba, Yauco	30-550	Canal, Mangrove, Stream	Forest, Grassland, Urban	
		Erythemis					
			plebeja (Burmeister, 1839)	Camuy, Coamo, Morovis, Ponce, Yauco	1-281	Pond, Lake / Reservoir, Stream	Agriculture, Forest, Grassland, Urban	
			vesiculosa (Fabricius, 1775)	Aguadilla, Añasco, Bayamón, Camuy, Carolina, Coamo, Guánica, Guayanilla, Humacao, Isabela, Jayuya, Juana Díaz, Mayagüez, Morovis, Ponce, Rincón, San Juan, Trujillo Alto, Yabucoa, Yauco	1-530	Canal, Open area, Pond, Stream	Agriculture, Forest, Grassland, Urban	
		Erythrodiplax					
			berenice (Drury, 1773)	Fajardo, Ponce, Santa Isabel, Vieques, Yabucoa	0-47	Lake/Reservoir, Mangrove, River mouth/Mangrove, Stream	Forest, Grassland, Urban	
			justiniana (Selys in Sagra, 1857)	Adjuntas, Carolina, Guayama, Guayanilla, Isabela, Jayuya, Lares, Las Piedras, Manatí, Maunabo, Morovis, Ponce, San Juan, Yauco	0-670	Canal, Lake/Reservoir, Pond, Stream	Agriculture, Forest, Grassland, Urban	
			umbrata (Linnaeus, 1758)	Adjuntas, Aguada, Aguadilla, Añasco, Arecibo, Barceloneta, Bayamón, Cabo Rojo, Caguas, Camuy, Carolina, Cayey, Ciales, Coamo, Corozal, Guánica, Guayama, Guayanilla, Gurabo, Hatillo, Humacao, Isabela, Jayuya, Juana Díaz, Lajas, Lares, Las Piedras, Loíza, Manatí, Maricao, Mayagüez, Morovis, Naguabo, Ponce, Rincón, Sabana Grande, Salinas, San Germán, San Juan, San Sebastián, Santa Isabel, Toa Alta, Trujillo Alto, Utuado, Vega Baja-Manatí, Yabucoa, Yauco	0- 840	Canal, Lake/Reservoir, Mangrove, Open Area, Pond, Stream, Wetland	Agriculture, Forest, Grassland, Urban	
		Idiataphe					
			cubensis (Scudder, 1866)	Manatíb	2	Lagoon	Grassland	
		Macrothemis					
			celeno (Selys in Sagra, 1857)	Adjuntas, Carolina, Ciales, Corozal, Guayanilla, Guaynabo, Humacao, Isabela, Jayuya, Juana Díaz, Lares, Maricao, Morovis, Peñuelas, Ponce, San Juan, Santa Isabel, Utuado, Yauco	0-1130	Canal, Lake/Reservoir, Mangrove, Stream, Wetland	Agriculture, Forest, Grassland, Urban	
		Miathyria					
			marcella (Selys in Sagra, 1857)	Aguada, Arecibo, Cabo Rojo, Caguas, Camuy, Carolina, Cidra, Juana Díaz, Lajas, Manatí, Mayagüez, Ponce, Utuado, Yabucoa, Yauco	0-477	Canal, Lake/Reservoir, Open Area, Pond, Stream, Wetland	Agriculture, Grassland, Urban	
			simplex (Rambur, 1842)	Manatíb	2	Lagoon	Grassland	
		Micrathyria					
			aequalis (Hagen, 1861)	Cabo Rojoe, Cabo Rojo, Camuy, Carolina, Corozal, Guánica, Manatí, Ponce, Yauco	0-281	Lake/Reservoir, Mangrove, Pond, Stream	Agriculture, Grassland, Urban	
			didyma (Selys in Sagra, 1857)	Loiza, Ponce, San Juan	0-22		Urban	
			dissocians Calvert, 1906	Aguada, Arecibo	3-7	Stream, Wetland	Agriculture	
		Orthemis					
			macrostigma (Rambur, 1842)	Aguadilla, Arecibo, Cabo Rojo, Camuy, Carolina, Coamo, Fajardo, Guánica, Guayanilla, Guaynabo, Humacao, Jayuya, Juana Díaz, Lares, Las Piedras, Mayagüez, Morovis, Ponce, Quebradillas, Río Grande, San Sebastián, Santa Isabel, Utuado, Vieques, Villalba, Yabucoa, Yauco	0-485	Canal, Estuary, Lake/Reservoir, Mangrove, Open Area, Pond, River mouth/Mangrove, Stream, Wetland	Agriculture, Forest, Grassland, Urban	
		Pantala					
			flavescens (Fabricius, 1798)	Aguadilla, Arecibo, Cabo Rojo, Caguas, Camuy, Carolina, Fajardo, Guánica, Guayanilla, Hormigueros, Humacao, Isabela, Juana Díaz, Peñuelas, Ponce, Salinas, San Juan, Santa Isabel, Vieques, Yauco	0-402	Canal, Lake/Reservoir, Mangrove, Open Area, River mouth/ Mangrove, Stream, Wetland	Agriculture, Forest, Grassland, Urban	
			hymenaea (Say, 1840)	Culebra	0	Open area	Beach	
		Perithemis					
			domitia (Drury, 1773)	Cabo Rojo, Camuy, Carolina, Ciales, Coamo, Humacao, Juana Díaz, Juncos, Lares, Las Piedras, Manatí, Maricao, Ponce, Río Grande, San Juan	0-456	Lake/Reservoir, Mangrove, Pond, Stream	Forest, Grassland, Urban	
		Scapanea					
			frontalis (Burmeister, 1839)	Adjuntas, Aibonito, Guayanilla, Jayuya, Maricao, Orocovis, Peñuelas, Ponce, Río Grande, Utuado, Villalba, Yauco	104-1,130	Stream	Forest, Grassland, Urban	
		Tauriphila					
			australis (Hagen, 1867)	Cidra, Lajas, Vega Bajab	4-400	Lagoon	Grassland	
		Tholymis					
			citrina (Hagen, 1867)	San Juan	21		Urban	
		Tramea					
			abdominalis (Rambur, 1842)	Aguadilla, Arecibo, Bayamón, Cabo Rojo, Carolina, Coamo, Fajardo, Guayanilla, Isabela, Juana Díaz, Mayagüez, Quebradillas, Santa Isabel, Yabucoa, Yauco	0-164	Canal, Estuary, Lake / Reservoir, Mangrove, Open Area, Pond, Stream	Agriculture, Forest, Grassland, Urban	
			binotata (Rambur, 1842)	Manatía				
			calverti Muttkowski, 1910	Cabo Rojoe, Guánica	39	Open area	Grassland	
			insularis Hagen, 1861	Manatíb	4	Lagoon	Grassland	
			onusta (Hagen, 1861)	Fajardo, Vieques	0-47	River mouth/ Mangrove	Forest	
Zygoptera					
	Coenagrionidae					
		Enallagma					
			civile (Hagen, 1861)	Arecibo, Yauco	7–37	Lake/ Reservoir, Stream	Agriculture	
			coecum (Hagen, 1861)	Adjuntas, Aguada, Aguadilla, Aibonito, Añasco, Arecibo, Barranquitas, Bayamón, Carolina, Ciales, Coamo, Corozal, Guánica, Guayama, Guayanilla, Guaynabo, Humacao, Jayuya, Juana Díaz, Juncos, Lares, Las Marías, Las Piedras, Maricao, Mayagüez, Moca, Morovis, Naguabo, Orocovis, Patillas, Peñuelas, Ponce, Rincón, Río Grande, Sabana Grande, San Germán, San Juan, San Sebastián, Santa Isabel, Utuado, Villalba, Yauco	0-750	Canal, Lake/ Reservoir, Mangrove, Open Area, Pond, Stream, Wetland	Forest, Grassland, Urban	
		Ischnura					
			hastata (Say, 1839)	Aguada, Aguadilla, Añasco, Arecibo, Bayamón, Carolina, Guánica, Lajas, Manatí, Maunabo, Moca, Ponce, San Juan, Santa Isabel	0-93	Canal, Estuary, Lake/ Reservoir, Mangrove, Open Area, Pond, Stream, Wetland	Agriculture, Forest, Grassland, Urban	
			ramburii (Selys, 1850)	Aguada, Aguadilla, Aibonito, Añasco, Arecibo, Barceloneta, Bayamón, Cabo Rojo, Caguas, Camuy, Carolina, Ciales, Cidra, Coamo, Corozal, Dorado, Fajardo, Guánica, Guayanilla, Guaynabo, Hatillo, Humacao, Isabela, Juana Díaz, Lajas, Manatí, Mayagüez, Naguabo, Ponce, Quebradillas, Salinas, San Juan, Santa Isabel, Toa Alta, Trujillo Alto, Utuado, Vega Baja-Manatí, Vieques, Villalba, Yabucoa, Yauco	0-525	Canal, Estuary, Lake/Reservoir, Mangrove, Open Area, Pond, River mouth/ Mangrove, Stream, Wetland	Agriculture, Forest, Grassland, Urban	
		Leptobasis					
			vacillans Hagen in Selys 1877	Cabo Rojoe, Juncos, Moca	14	Canal	Grassland	
		Nehalennia					
			minuta (Selys in Sagra, 1857)	Moca			Grassland	
		Neoerythromma					
			cultellatum (Hagen in Selys, 1876)	Arecibo, Bayamón, Carolina, Manatí, San Juan, Yauco	1-138	Lake/ Reservoir, Mangrove, Stream	Grassland, Urban	
		Protoneura					
			viridis Westfall, 1964	Aibonito, Coamo, Guayanilla, Hatillo, Mayagüez, Morovis, Peñuelas, Ponce, San Germán, San Juan	1-660	Lake/Reservoir, Stream	Forest, Grassland, Urban	
		Telebasis					
			dominicana (Selys in Sagra, 1857)	Aguada, Aibonito, Arecibo, Bayamón, Cabo Rojo, Camuy, Carolina, Cayey, Coamo, Hatillo, Jayuya, Juana Díaz, Juncos, Lajas, Manatí, Mayagüez, Moca, Naguabo, Ponce, Río Grande, San Germán, San Juan, Vega Baja-Manatí, Villalba, Yauco	0-1130	Canal, Lake/Reservoir, Mangrove, Open area, Pond, Stream, Wetland	Agriculture, Forest, Grassland, Urban	
			vulnerata (Hagen, 1861)	Adjuntas, Aibonito, Barranquitas, Bayamón, Caguas, Carolina, Corozal, Guayanilla, Guaynabo, Jayuya, Juana Díaz, Juncos, Las Piedras, Maricao, Mayagüez, Moca, Morovis, Naguabo, Orocovis, Patillas, Peñuelas, Ponce, Río Grande, San Germán, San Juan, Utuado, Vega Baja-Manatí, Villalba, Yabucoa	4-700	Canal, Lake/Reservoir, Open Area, Stream, Wetland	Agriculture, Forest, Grassland, Urban	
	Lestidae					
		Lestes					
			forficula Rambur, 1842	Manatíb	4	Lagoon	Grassland	
			scalaris Gundlach, 1888	Floridad				
			spumarius Hagen in Selys, 1862	Arecibo, Guánica	10	Wetland	Grassland	
Notes.

a Reported by Klots (1932).

b R Garrison, pers. comm., 2019.

c Reported by Fliedner (2009).

d Reported by García Díaz (1938).

e D Paulson, pers. comm., 2020.

Our survey covered the main island and some of the smaller islands adjacent (e.g., Vieques, Culebra) and sampling was conducted at all elevations (Table 1). Odonata were found inhabiting all water bodies sampled (e.g., streams, ponds, lagoons, estuaries) and all land uses (e.g., forest, grasslands, urban areas; Table 1).

None of the species in our list for Puerto Rico is endemic to the island, but several are endemic to the Caribbean. Three species of Libellulidae (Erythrodiplax justiniana, Macrothemis celeno, and Scapanea frontalis) and four of Coenagrionidae (Telebasis dominicana, T. vulnerata, Lestes scalaris, and Protoneura viridis) are Caribbean endemics. Among the most abundant species of Zygoptera were Telebasis vulnerata and Ischnura ramburii (Coenagrionidae) (Fig. 2) and the Anisoptera Dythemis rufinervis, Erythemis vesiculosa, and Orthemis cf. macrostigma (Libellulidae) (Fig. 3).

Table 2 Species recorded in Puerto Rico during the 1930s –1940s, species deposited at MZUPR, and information on first reports to the island.

				Pre-1940		Specimen at	First	
		Species	K+GD		MZUPR	MZUPR	report	
Anisoptera						
	Aeshnidae						
		Anax amazili	x		x	x	García Díaz (1938)	
		Anax junius	x		x	x	Klots (1932)	
		Coryphaeschna adnexa	x			x	Klots (1932)	
		Coryphaeschna viriditas				x	This study	
		Gynacantha nervosa	x		x	x	Kolbe (1888)	
		Rhionaeschna psilus					Von Ellenrieder (2003)	
		Triacanthagyna caribbea				x	Von Ellenrieder & Garrison (2003)	
		Triacanthagyna septima			x	x	Von Ellenrieder & Garrison (2003)	
	Libellulidae						
		Brachymesia furcata	x			x	Wolcott (1948)	
		Brachymesia herbida	x			x	Klots (1932)	
		Crocothemis serviliaa					Fliedner (2009)	
		Dythemis rufinervis	x		x	x	Kolbe (1888)	
		Erythemis plebeja	x			x	Klots (1932)	
		Erythemis vesiculosa	x		x	x	Kolbe (1888)	
		Erythrodiplax berenice	x		x	x	Klots (1932)	
		Erythrodiplax justiniana	x			x	Kolbe (1888)	
		Erythrodiplax umbrata	x		x	x	Kolbe (1888)	
		Idiataphe cubensis	x				García Díaz (1938)	
		Macrothemis celeno	x		x	x	Kolbe (1888)	
		Miathyria marcella	x		x	x	Klots (1932)	
		Miathyria simplex					This study, R Garrison, pers. comm., 2019	
		Micrathyria aequalis	x			x	García Díaz (1938)	
		Micrathyria didyma	x			x	Kolbe (1888)	
		Micrathyria dissocians	x			x	García Díaz (1938)	
		Orthemis macrostigma	x		x	x	Kolbe (1888)	
		Pantala flavescens	x		x	x	Kolbe (1888)	
		Pantala hymenaea					This study	
		Perithemis domitia	x		x	x	Kolbe (1888)	
		Scapanea frontalis	x		x	x	Klots (1932)	
		Tauriphila australis				x	This study, R Garrison, pers. comm., 2019	
		Tholymis citrina					This study	
		Tramea abdominalis	x			x	Kolbe (1888)	
		Tramea binotata	x				Klots (1932)	
		Tramea calverti				x	This study	
		Tramea insularis					This study, R Garrison, pers. comm., 2019	
		Tramea onusta	x			x	García Díaz (1938)	
Zygoptera						
	Coenagrionidae						
		Enallagma civile	x			x	Kolbe (1888)	
		Enallagma coecum	x			x	Kolbe (1888)	
		Ischnura hastata	x			x	García Díaz (1938)	
		Ischnura ramburii	x		x	x	Kolbe (1888)	
		Leptobasis vacillans	x			x	Kolbe (1888)	
		Nehalennia minuta	x			x	García Díaz (1938)	
		Neoerythromma cultellatum	x			x	García Díaz (1938)	
		Protoneura viridis	x		x	x	Westfall Jr (1964)	
		Telebasis dominicana	x			x	García Díaz (1938)	
		Telebasis vulnerata	x		x	x	Gundlach (1893)	
	Lestidae						
		Lestes forficula	x				Klots (1932)	
		Lestes scalaris	x				Klots (1932)	
		Lestes spumarius	x				Kolbe (1888)	
Notes.

a Introduced species.

Our list of records from the 1930s–1940s resulted in 39 species for Puerto Rico (Table 2). The difference between lists includes some elusive species, like Tholymis citrina, and strong fliers that are often difficult to catch (e.g., Coryphaeschna viriditas).

Family overview

Aeshnidae (Suborder Anisoptera)

This family has five genera and eight species on the island (Table 1). Collections mostly occurred at low elevations, with some specimens of Triacanthagyna collected at mid elevations (∼400 m). Species in this family have large bodies and are strong fliers. We found them in all land uses, often flying during sundown hours. The family include groups with broad distributions, like Anax, a cosmopolitan genus.

Figure 2 Common species of Zygoptera.

Two abundant species of Coenagrionidae: (A) Telebasis vulnerata (Photo Alonso Ramírez) and (B) Ischnura ramburii (Photo Norman Maldonado-Benítez).

Figure 3 Common species of Anisoptera.

Three abundant Libellulidae: (A) Dythemis rufinervis (Photo Norman Maldonado-Benítez), (B) Erythemis vesiculosa (Photo Norman Maldonado-Benítez), and (C) Orthemis cf. macrostigma (Photo Alonso Ramírez).

Libellulidae (Suborder Anisoptera)

This family has 15 genera and 27 species, making it the largest group on the island (Table 1). We found them all over the island and at all elevations and land uses, but more diverse in lowlands. Erythrodiplax umbrata was the most common species in our survey, abundant in open areas. The most elusive species was perhaps Tholymis citrina, we found only one specimen in our survey. The family also included cosmopolitan species (e.g., Pantala flavescens) and Caribbean endemics (Scapanea frontalis). The introduced species Crocothemis servilia was reported for Puerto Rico (Fliedner, 2009), but was not found during our survey.

The identity of Orthemis species in Puerto Rico remains unresolved. O. macrostigma has been reported for the Caribbean (Meurgey, 2013) along with another morph; often called the “Antillean red” (D Paulson, pers. comm., 2020).

Coenagrionidae (Suborder Zygoptera)

This is the second largest family, with seven genera and 10 species (Table 1). It is well distributed around the island and in all land uses and elevations. The most common species were Enallagma coecum, Ischnura hastata and I. ramburii, and Telebasis vulnerata. The latter species a Caribbean endemic.

Lestidae (Suborder Zygoptera)

This is the smallest family in Puerto Rico, with only three species (Table 1). Lestids specialize in wetland and pond environments with abundant vegetation and might be able to deal with some degree of disturbance. We found few specimens and all from low elevations sites.

Discussion

Our study reports the occurrence of 49 species of Odonata in Puerto Rico. We are confident that this is a good representation of the Puerto Rican fauna, as we have specimens for 41 of the species and the remaining eight had been reported by reliable sources. Previous studies reported fewer species. The earliest and most comprehensive list of species for Puerto Rico can be compiled by combining reports by Klots (1932), García Díaz (1938), which reported a combined total of 39 species. The next comprehensive report for Puerto Rico was published by Meurgey (2013) in his catalogue of West Indian dragonflies, but his list for Puerto Rico includes only 36 species. Our coverage was extensive and most municipalities in Puerto Rico were visited. Species were recorded from all elevations, sea level to mountains, and in all types of habitats, including coastal ecosystems (e.g., mangroves and estuaries).

The total number of species in Puerto Rico compares well with those from the remaining West Indies islands. Puerto Rico is the smallest of the Greater Antilles and its biodiversity follows the same pattern, with Cuba, Jamaica, and Hispaniola all having higher diversity than Puerto Rico (Ackerman, Trejo-Torres & Crespo-Chuy, 2007). According to Meurgey (2013), Cuba has 88 species, Jamaica 58, Haiti 58, and the Dominican Republic 67. The Lesser Antilles are all much smaller in area than Puerto Rico and have fewer species, with Guadalupe as the most diverse with 38 species (Meurgey, 2013). While additional reports for Puerto Rico are possible, we do not expect many more species to be added to the current inventory of 48 species.

Puerto Rico was dominated by agricultural activities until the mid-1940s, when the island transitioned to an industrial based economy (Grau et al., 2003). Agriculture and deforestation reached a maximum in the 1940s. The increase in industry jobs resulted in the abandonment of marginal agricultural lands and eventual natural reforestation. Today, Puerto Rico is one of the few tropical locations that has gained forest, rather than suffering from deforestation (Aide et al., 2013). Earliest Odonata reports for the island were compiled during the period of maximum agriculture (>90% of the island). The combined list of Odonata taxa for the period is of 39 species, which is over 80% of the species currently reported. The species we are adding to that historic list include habitat generalist and strong fliers (e.g., Pantala hymenaea), with a few exceptions (e.g., Tholymis citrina). Thus, the species list of Puerto Rico today mostly reflects that of the agricultural period. This contrasts with the rest of the Greater Antilles that have species that are forest specialists and island endemics. It is possible that the size of Puerto Rico never allowed for higher diversification or that the island lost part of its fauna before we were able to document their occurrence.

Future threats to freshwater ecosystems and Odonata in Puerto Rico and the Caribbean will come with the changing climate in the region. Climate change models predict an increase in drought frequency and less overall precipitation over the Caribbean (Van Beusekom et al., 2016). Additionally, more frequent and severe storms are expected with changing climate conditions. Conservation efforts should focus on assessing the implications of drought on habitat availability for odonates. It is also important to understand how changes in climate (e.g., precipitation, temperature) might affect different species. We provide this list as a first step toward improving our understanding and their conservation of Odonata in the island.

Conclusions

Freshwater ecosystems in the Caribbean are under strong pressure from anthropogenic activities and climate change. Our study provides based line information advancing our understanding of freshwater biodiversity in Puerto Rico, the smallest of the Greater Antilles. Our report of 49 species for the island corresponds well with it size and position in the Caribbean. In addition, it is similar to original reports from before the 1940s, indicating that Puerto Rico freshwater diversity might reflect the land use legacy of an intense agricultural period. Complete and reliable information on freshwater biodiversity is key to understand and manage aquatic ecosystems in the face of predicted changes resulting from climate change.

Our survey of Odonata in Puerto Rico was possible thanks to the support of the Puerto Rico Louis Stokes Alliance for Minority Participation and the participants of the Dragonfly Project. We also appreciate the support of Rosser Garrison and Natalia von Ellenrieder who provided literature and support for our project and Lucía Ramírez for her support in preparing the tables for the publication.

Additional Information and Declarations

Competing Interests

Author Contributions

Data Availability

The authors declare there are no competing interests.

Alonso Ramírez conceived and designed the experiments, performed the experiments, analyzed the data, prepared figures and/or tables, authored or reviewed drafts of the paper, and approved the final draft.

Norman Maldonado-Benítez performed the experiments, analyzed the data, prepared figures and/or tables, authored or reviewed drafts of the paper, and approved the final draft.

Ashley Mariani-Ríos performed the experiments, analyzed the data, authored or reviewed drafts of the paper, and approved the final draft.

Javier Figueroa-Santiago conceived and designed the experiments, performed the experiments, authored or reviewed drafts of the paper, and approved the final draft.

The following information was supplied regarding data availability:

Data is available at: Ramirez, Alonso. (2020). A survey of Odonata of Puerto Rico [Data set]. Zenodo. http://doi.org/10.5281/zenodo.3841637.

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
