# Peer review of "Dragonflies and damselflies (Odonata) from Puerto Rico: a checklist with notes on distribution and habitat"

_PeerJ, doi:10.7717/peerj.9711_

## Round 0.1 · original submission · Major Revisions

Dear Dr. Ramírez and colleagues:

Thanks for submitting your manuscript to PeerJ. I have now received three independent reviews of your work, and as you will see, the reviewers raised some concerns about the research. Despite this, these reviewers are optimistic about your work and the potential impact it will have on research studying odonates of Puerto Rico. Thus, I encourage you to revise your manuscript, accordingly, taking into account all of the concerns raised by the three reviewers.

Please add all appropriate references that should make the study inclusive.

Please note that Reviewer 2 kindly provided a marked-up version of your manuscript.

Please also ensure that your figures contain all of the information that is necessary to support your findings and observations. Captions appear to have errors. Please revise the map in Figure 1 with a focus on PR.

While the concerns of the reviewers are relatively minor, this is a major revision to ensure that the original reviewers have a chance to evaluate your responses to their concerns. There are many suggestions, which I am sure will greatly improve your manuscript once addressed.

Good luck with your revision,

-joe

Reviewer 1 ·

Basic reporting

The English language is used properly,
Only few references are listed, which does not reflect the ambition to characterize the dragonfly fauna of a large area.
The article is well structured, but it remains purely descriptive. There is not hypotheses that can be tested. This would mean that the standards for research papers are not met. This cannot be solved by improvements, as the paper is a pure list of recorded species.
However, to my opinion, it is a merit to publish such a list of species and to make this knowledge available to the scientific community. The question is just whether a journal such as PeerJ is the right platform for this.
The figure 1 is not helpful as it does not show the investigation area but rather the Carribean archipelago.
Figure captions of figure 2 and 3 are mixed up.
The only contents-oriented figures are those 5 fotos of dragonflies.
The major quality of the manuscript is in the tables of the appendix.

Experimental design

There is no experiment. The sampling is not well designed, as the paper concludes the result of almost 20 years of field work and recording. I do have concern of the data quality for the spatial distribution of specimen accross the island, as field work was done to a great part by unexperienced undergraduate students. The aim was obviously, together with expert opinions, to creat a list of Odonata species. This is a very important work, but it does not satisfy the standards of a peer-reviewed research paper. Such data should rather be made available in data respositories for further large-scale analyses.

Validity of the findings

As the data were recorded over a long period and data sources are inconsistent, i do see only a limited validity of the findings because it will be difficult to yield sound results for changes in the future based on this species list.

Additional comments

I appreciate your work but i recommend to publish these data rather in data repositories than in research journals.

·

Basic reporting

no comment

Experimental design

no comment; the paper is descriptive, with no experimental design

Validity of the findings

no comment

Additional comments

Word document attached with all my comments

·

Basic reporting

General:

1) The manuscript is clearly written in unambiguous language
2) It has a professional structure
3) The literature is appropriate as support and context according to the scope of the ms
4) Tables are clear with abundant information


-Figure 1: I strongly recommend to improve this map, it highlights Dominican Republic instead of Puerto Rico, and could be confuse for readers from other parts of the World.

Experimental design

Specific:

-Introduction: Lines 87-88, please provide a couple of examples of groups with major gaps;
Lines 120-122, it would be interesting if you could discuss if the Meurgey’s reduction in the number of species for PR is according with your findings.

-Materials & Methods: Lines 148-151, the surveys were sporadic, in a monthly fashion or focused in a particular season?

Validity of the findings

-Results: Lines 173-176, please provide the citations (author, date) whose records were omitted from your list. This would be useful if in the future these species are found.

-Discussion: You found 9 species more than those of the historical records, could be due to a major collecting effort, immigrations, or anything else? I know this could fall in speculative grounds but maybe you could have some ideas to this respect.

-References: Lines 337-340, these citations should be inverted.

Additional comments

I congratulate to the authors for this interesting contribution that up to date the list of Odonata from Puerto Rico. It is important in the context of climate change that is enhancing the colonization of new areas by several species of dragonflies and damselflies.

---

## Round 0.2 · accepted · Accept

Dear Dr. Ramírez and colleagues:

Thanks for revising your manuscript based on the concerns raised by the reviewers. I now believe that your manuscript is suitable for publication. Congratulations! I look forward to seeing this work in print, and I anticipate it being an important resource for groups studying odonates of Puerto Rico. Thanks again for choosing PeerJ to publish such important work.

Best,

-joe

·

Basic reporting

.

Experimental design

.

Validity of the findings

.

Additional comments

This was a second review to confirm that the authors had responded to questions and comments on the first review. They did so to my satisfaction.